# COMBINING POLICY GRADIENT AND Q-LEARNING

**Brendan O'Donoghue, Rémi Munos, Koray Kavukcuoglu & Volodymyr Mnih**
Deepmind
{bodonoghue,munos,korayk,vmnih}@google.com

## ABSTRACT

Policy gradient is an efficient technique for improving a policy in a reinforcement learning setting. However, vanilla online variants are on-policy only and not able to take advantage of off-policy data. In this paper we describe a new technique that combines policy gradient with off-policy Q-learning, drawing experience from a replay buffer. This is motivated by making a connection between the fixed points of the regularized policy gradient algorithm and the Q-values. This connection allows us to estimate the Q-values from the action preferences of the policy, to which we apply Q-learning updates. We refer to the new technique as 'PGQL', for policy gradient and Q-learning. We also establish an equivalency between action-value fitting techniques and actor-critic algorithms, showing that regularized policy gradient techniques can be interpreted as advantage function learning algorithms. We conclude with some numerical examples that demonstrate improved data efficiency and stability of PGQL. In particular, we tested PGQL on the full suite of Atari games and achieved performance exceeding that of both asynchronous advantage actor-critic (A3C) and Q-learning.

## 1 INTRODUCTION

In reinforcement learning an agent explores an environment and through the use of a reward signal learns to optimize its behavior to maximize the expected long-term return. Reinforcement learning has seen success in several areas including robotics (Lin, 1993; Levine et al., 2015), computer games (Mnih et al., 2013; 2015), online advertising (Pednault et al., 2002), board games (Tesauro, 1995; Silver et al., 2016), and many others. For an introduction to reinforcement learning we refer to the classic text by Sutton & Barto (1998). In this paper we consider model-free reinforcement learning, where the state-transition function is not known or learned. There are many different algorithms for model-free reinforcement learning, but most fall into one of two families: action-value fitting and policy gradient techniques.

Action-value techniques involve fitting a function, called the Q-values, that captures the expected return for taking a particular action at a particular state, and then following a particular policy thereafter. Two alternatives we discuss in this paper are SARSA (Rummery & Niranjan, 1994) and Q-learning (Watkins, 1989), although there are many others. SARSA is an on-policy algorithm whereby the action-value function is fit to the current policy, which is then refined by being mostly greedy with respect to those action-values. On the other hand, Q-learning attempts to find the Q-values associated with the optimal policy directly and does not fit to the policy that was used to generate the data. Q-learning is an off-policy algorithm that can use data generated by another agent or from a replay buffer of old experience. Under certain conditions both SARSA and Q-learning can be shown to converge to the optimal Q-values, from which we can derive the optimal policy (Sutton, 1988; Bertsekas & Tsitsiklis, 1996).

In policy gradient techniques the policy is represented explicitly and we improve the policy by updating the parameters in the direction of the gradient of the performance (Sutton et al., 1999; Silver et al., 2014; Kakade, 2001). Online policy gradient typically requires an estimate of the action-value function of the current policy. For this reason they are often referred to as actor-critic methods, where the actor refers to the policy and the critic to the estimate of the action-value function (Konda & Tsitsiklis, 2003). Vanilla actor-critic methods are on-policy only, although some attempts have been made to extend them to off-policy data (Degris et al., 2012; Levine & Koltun, 2013).

In this paper we derive a link between the Q-values induced by a policy and the policy itself when the policy is the fixed point of a regularized policy gradient algorithm (where the gradient vanishes). This connection allows us to derive an estimate of the Q-values from the current policy, which we can refine using off-policy data and Q-learning. We show in the tabular setting that when the regularization penalty is small (the usual case) the resulting policy is close to the policy that would be found without the addition of the Q-learning update. Separately, we show that regularized actor-critic methods can be interpreted as action-value fitting methods, where the Q-values have been parameterized in a particular way. We conclude with some numerical examples that provide empirical evidence of improved data efficiency and stability of PGQL.

## 1.1 PRIOR WORK

Here we highlight various axes along which our work can be compared to others. In this paper we use entropy regularization to ensure exploration in the policy, which is a common practice in policy gradient (Williams & Peng, 1991; Mnih et al., 2016). An alternative is to use KL-divergence instead of entropy as a regularizer, or as a constraint on how much deviation is permitted from a prior policy (Bagnell & Schneider, 2003; Peters et al., 2010; Schulman et al., 2015; Fox et al., 2015). Natural policy gradient can also be interpreted as putting a constraint on the KL-divergence at each step of the policy improvement (Amari, 1998; Kakade, 2001; Pascanu & Bengio, 2013). In Sallans & Hinton (2004) the authors use a Boltzmann exploration policy over estimated Q-values which they update using TD-learning. In Heess et al. (2012) this was extended to use an actor-critic algorithm instead of TD-learning, however the two updates were not combined as we have done in this paper. In Azar et al. (2012) the authors develop an algorithm called dynamic policy programming, whereby they apply a Bellman-like update to the action-preferences of a policy, which is similar in spirit to the update we describe here. In Norouzi et al. (2016) the authors augment a maximum likelihood objective with a reward in a supervised learning setting, and develop a connection that resembles the one we develop here between the policy and the Q-values. Other works have attempted to combine on and off-policy learning, primarily using action-value fitting methods (Wang et al., 2013; Hausknecht & Stone, 2016; Lehnert & Precup, 2015), with varying degrees of success. In this paper we establish a connection between actor-critic algorithms and action-value learning algorithms. In particular we show that TD-actor-critic (Konda & Tsitsiklis, 2003) is equivalent to expected-SARSA (Sutton & Barto, 1998, Exercise 6.10) with Boltzmann exploration where the Q-values are decomposed into advantage function and value function. The algorithm we develop extends actor-critic with a Q-learning style update that, due to the decomposition of the Q-values, resembles the update of the dueling architecture (Wang et al., 2016). Recently, the field of deep reinforcement learning, *i.e.*, the use of deep neural networks to represent action-values or a policy, has seen a lot of success (Mnih et al., 2015; 2016; Silver et al., 2016; Riedmiller, 2005; Lillicrap et al., 2015; Van Hasselt et al., 2016). In the examples section we use a neural network with PGQL to play the Atari games suite.

## 2 REINFORCEMENT LEARNING

We consider the infinite horizon, discounted, finite state and action space Markov decision process, with state space $\mathcal{S}$, action space $\mathcal{A}$ and rewards at each time period denoted by $r_t \in \mathbb{R}$. A *policy* $\pi : \mathcal{S} \times \mathcal{A} \to \mathbb{R}_+$ is a mapping from state-action pair to the probability of taking that action at that state, so it must satisfy $\sum_{a \in \mathcal{A}} \pi(s, a) = 1$ for all states $s \in \mathcal{S}$. Any policy $\pi$ induces a probability distribution over visited states, $d^\pi : \mathcal{S} \to \mathbb{R}_+$ (which may depend on the initial state), so the probability of seeing state-action pair $(s, a) \in \mathcal{S} \times \mathcal{A}$ is $d^\pi(s)\pi(s, a)$.

In reinforcement learning an 'agent' interacts with an environment over a number of times steps. At each time step $t$ the agent receives a state $s_t$ and a reward $r_t$ and selects an action $a_t$ from the policy $\pi_t$, at which point the agent moves to the next state $s_{t+1} \sim P(\cdot, s_t, a_t)$, where $P(s', s, a)$ is the probability of transitioning from state $s$ to state $s'$ after taking action $a$. This continues until the agent encounters a terminal state (after which the process is typically restarted). The goal of the agent is to find a policy $\pi$ that maximizes the expected total discounted return $J(\pi) = \mathbf{E}(\sum_{t=0}^\infty \gamma^t r_t \mid \pi)$, where the expectation is with respect to the initial state distribution, the state-transition probabilities, and the policy, and where $\gamma \in (0, 1)$ is the discount factor that, loosely speaking, controls how much the agent prioritizes long-term versus short-term rewards. Since the agent starts with no knowledge

of the environment it must continually explore the state space and so will typically use a stochastic policy.

**Action-values.** The action-value, or Q-value, of a particular state under policy $\pi$ is the expected total discounted return from taking that action at that state and following $\pi$ thereafter, *i.e.*, $Q^\pi(s,a) = \mathbf{E}(\sum_{t=0}^\infty \gamma^t r_t \mid s_0 = s, a_0 = a, \pi)$. The value of state $s$ under policy $\pi$ is denoted by $V^\pi(s) = \mathbf{E}(\sum_{t=0}^\infty \gamma^t r_t \mid s_0 = s, \pi)$, which is the expected total discounted return of policy $\pi$ from state $s$. The optimal action-value function is denoted $Q^\star$ and satisfies $Q^\star(s,a) = \max_\pi Q^\pi(s,a)$ for each $(s,a)$. The policy that achieves the maximum is the optimal policy $\pi^\star$, with value function $V^\star$. The advantage function is the difference between the action-value and the value function, *i.e.*, $A^\pi(s,a) = Q^\pi(s,a) - V^\pi(s)$, and represents the additional expected reward of taking action $a$ over the average performance of the policy from state $s$. Since $V^\pi(s) = \sum_a \pi(s,a)Q^\pi(s,a)$ we have the identity $\sum_a \pi(s,a)A^\pi(s,a) = 0$, which simply states that the policy $\pi$ has no advantage over itself.

**Bellman equation.** The Bellman operator $\mathcal{T}^\pi$ (Bellman, 1957) for policy $\pi$ is defined as

$$\mathcal{T}^\pi Q(s,a) = \mathop{\mathbf{E}}_{s',r,b}(r(s,a) + \gamma Q(s',b)),$$

where the expectation is over next state $s' \sim P(\cdot, s, a)$, the reward $r(s,a)$, and the action $b$ from policy $\pi_{s'}$. The Q-value function for policy $\pi$ is the fixed point of the Bellman operator for $\pi$, *i.e.*, $\mathcal{T}^\pi Q^\pi = Q^\pi$. The optimal Bellman operator $\mathcal{T}^\star$ is defined as

$$\mathcal{T}^\star Q(s,a) = \mathop{\mathbf{E}}_{s',r}(r(s,a) + \gamma \max_b Q(s',b)),$$

where the expectation is over the next state $s' \sim P(\cdot, s, a)$, and the reward $r(s,a)$. The optimal Q-value function is the fixed point of the optimal Bellman equation, *i.e.*, $\mathcal{T}^\star Q^\star = Q^\star$. Both the $\pi$-Bellman operator and the optimal Bellman operator are $\gamma$-contraction mappings in the sup-norm, *i.e.*, $\|\mathcal{T}Q_1 - \mathcal{T}Q_2\|_\infty \le \gamma\|Q_1 - Q_2\|_\infty$, for any $Q_1, Q_2 \in \mathbb{R}^{\mathcal{S}\times\mathcal{A}}$. From this fact one can show that the fixed point of each operator is unique, and that value iteration converges, *i.e.*, $(\mathcal{T}^\pi)^k Q \to Q^\pi$ and $(\mathcal{T}^\star)^k Q \to Q^\star$ from any initial $Q$. (Bertsekas, 2005).

## 2.1 ACTION-VALUE LEARNING

In value based reinforcement learning we approximate the Q-values using a function approximator. We then update the parameters so that the Q-values are as close to the fixed point of a Bellman equation as possible. If we denote by $Q(s,a;\theta)$ the approximate Q-values parameterized by $\theta$, then Q-learning updates the Q-values along direction $\mathbf{E}_{s,a}(\mathcal{T}^\star Q(s,a;\theta) - Q(s,a;\theta))\nabla_\theta Q(s,a;\theta)$ and SARSA updates the Q-values along direction $\mathbf{E}_{s,a}(\mathcal{T}^\pi Q(s,a;\theta) - Q(s,a;\theta))\nabla_\theta Q(s,a;\theta)$. In the online setting the Bellman operator is approximated by sampling and bootstrapping, whereby the Q-values at any state are updated using the Q-values from the next visited state. Exploration is achieved by not always taking the action with the highest Q-value at each time step. One common technique called 'epsilon greedy' is to sample a random action with probability $\epsilon > 0$, where $\epsilon$ starts high and decreases over time. Another popular technique is 'Boltzmann exploration', where the policy is given by the softmax over the Q-values with a temperature $T$, *i.e.*, $\pi(s,a) = \exp(Q(s,a)/T)/\sum_b \exp(Q(s,b)/T)$, where it is common to decrease the temperature over time.

## 2.2 POLICY GRADIENT

Alternatively, we can parameterize the policy directly and attempt to improve it via gradient ascent on the performance $J$. The policy gradient theorem (Sutton et al., 1999) states that the gradient of $J$ with respect to the parameters of the policy is given by

$$\nabla_\theta J(\pi) = \mathop{\mathbf{E}}_{s,a} Q^\pi(s,a)\nabla_\theta \log \pi(s,a), \tag{1}$$

where the expectation is over $(s,a)$ with probability $d^\pi(s)\pi(s,a)$. In the original derivation of the policy gradient theorem the expectation is over the *discounted* distribution of states, *i.e.*, over $d_\gamma^{\pi,s_0}(s) = \sum_{t=0}^\infty \gamma^t Pr\{s_t = s \mid s_0, \pi\}$. However, the gradient update in that case will assign a low

weight to states that take a long time to reach and can therefore have poor empirical performance. In practice the non-discounted distribution of states is frequently used instead. In certain cases this is equivalent to maximizing the average (*i.e.*, non-discounted) policy performance, even when $Q^\pi$ uses a discount factor (Thomas, 2014). Throughout this paper we will use the *non-discounted* distribution of states.

In the online case it is common to add an entropy regularizer to the gradient in order to prevent the policy becoming deterministic. This ensures that the agent will explore continually. In that case the (batch) update becomes

$$\Delta\theta \propto \mathop{\mathbf{E}}_{s,a} Q^\pi(s,a)\nabla_\theta \log \pi(s,a) + \alpha \mathop{\mathbf{E}}_{s} \nabla_\theta H^\pi(s), \tag{2}$$

where $H^\pi(s) = -\sum_a \pi(s,a)\log\pi(s,a)$ denotes the entropy of policy $\pi$, and $\alpha > 0$ is the regularization penalty parameter. Throughout this paper we will make use of entropy regularization, however many of the results are true for other choices of regularizers with only minor modification, *e.g.*, KL-divergence. Note that equation (2) requires exact knowledge of the Q-values. In practice they can be estimated, *e.g.*, by the sum of discounted rewards along an observed trajectory (Williams, 1992), and the policy gradient will still perform well (Konda & Tsitsiklis, 2003).

## 3 REGULARIZED POLICY GRADIENT ALGORITHM

In this section we derive a relationship between the policy and the Q-values when using a regularized policy gradient algorithm. This allows us to transform a policy into an estimate of the Q-values. We then show that for small regularization the Q-values induced by the policy at the fixed point of the algorithm have a small Bellman error in the tabular case.

### 3.1 TABULAR CASE

Consider the fixed points of the entropy regularized policy gradient update (2). Let us define $f(\theta) = \mathbf{E}_{s,a} Q^\pi(s,a)\nabla_\theta \log\pi(s,a) + \alpha \mathbf{E}_s \nabla_\theta H(\pi_s)$, and $g_s(\pi) = \sum_a \pi(s,a)$ for each $s$. A fixed point is one where we can no longer update $\theta$ in the direction of $f(\theta)$ without violating one of the constraints $g_s(\pi) = 1$, *i.e.*, where $f(\theta)$ is in the span of the vectors $\{\nabla_\theta g_s(\pi)\}$. In other words, any fixed point must satisfy $f(\theta) = \sum_s \lambda_s \nabla_\theta g_s(\pi)$, where for each $s$ the Lagrange multiplier $\lambda_s \in \mathbb{R}$ ensures that $g_s(\pi) = 1$. Substituting in terms to this equation we obtain

$$\mathop{\mathbf{E}}_{s,a}\left(Q^\pi(s,a) - \alpha\log\pi(s,a) - c_s\right)\nabla_\theta\log\pi(s,a) = 0, \tag{3}$$

where we have absorbed all constants into $c \in \mathbb{R}^{|\mathcal{S}|}$. Any solution $\pi$ to this equation is strictly positive element-wise since it must lie in the domain of the entropy function. In the tabular case $\pi$ is represented by a single number for each state and action pair and the gradient of the policy with respect to the parameters is the indicator function, *i.e.*, $\nabla_{\theta(t,b)}\pi(s,a) = \mathbf{1}_{(t,b)=(s,a)}$. From this we obtain $Q^\pi(s,a) - \alpha\log\pi(s,a) - c_s = 0$ for each $s$ (assuming that the measure $d^\pi(s) > 0$). Multiplying by $\pi(a,s)$ and summing over $a \in \mathcal{A}$ we get $c_s = \alpha H^\pi(s) + V^\pi(s)$. Substituting $c$ into equation (3) we have the following formulation for the policy:

$$\pi(s,a) = \exp(A^\pi(s,a)/\alpha - H^\pi(s)), \tag{4}$$

for all $s \in \mathcal{S}$ and $a \in \mathcal{A}$. In other words, the policy at the fixed point is a softmax over the advantage function induced by that policy, where the regularization parameter $\alpha$ can be interpreted as the temperature. Therefore, we can use the policy to derive an estimate of the Q-values,

$$\tilde{Q}^\pi(s,a) = \tilde{A}^\pi(s,a) + V^\pi(s) = \alpha(\log\pi(s,a) + H^\pi(s)) + V^\pi(s). \tag{5}$$

With this we can rewrite the gradient update (2) as

$$\Delta\theta \propto \mathop{\mathbf{E}}_{s,a}(Q^\pi(s,a) - \tilde{Q}^\pi(s,a))\nabla_\theta\log\pi(s,a), \tag{6}$$

since the update is unchanged by per-state constant offsets. When the policy is parameterized as a softmax, *i.e.*, $\pi(s,a) = \exp(W(s,a))/\sum_b \exp W(s,b)$, the quantity $W$ is sometimes referred to as the action-preferences of the policy (Sutton & Barto, 1998, Chapter 6.6). Equation (4) states that the action preferences are equal to the Q-values scaled by $1/\alpha$, up to an additive per-state constant.

## 3.2  GENERAL CASE

Consider the following optimization problem:

$$\begin{array}{ll} \text{minimize} & \mathbf{E}_{s,a}(q(s,a) - \alpha \log \pi(s,a))^2 \\ \text{subject to} & \sum_a \pi(s,a) = 1, \quad s \in \mathcal{S} \end{array} \tag{7}$$

over variable $\theta$ which parameterizes $\pi$, where we consider both the measure in the expectation and the values $q(s,a)$ to be independent of $\theta$. The optimality condition for this problem is

$$\underset{s,a}{\mathbf{E}}\,(q(s,a) - \alpha \log \pi(s,a) + c_s)\nabla_\theta \log \pi(s,a) = 0,$$

where $c \in \mathbb{R}^{|\mathcal{S}|}$ is the Lagrange multiplier associated with the constraint that the policy sum to one at each state. Comparing this to equation (3), we see that if $q = Q^\pi$ and the measure in the expectation is the same then they describe the same set of fixed points. This suggests an interpretation of the fixed points of the regularized policy gradient as a regression of the log-policy onto the Q-values. In the general case of using an approximation architecture we can interpret equation (3) as indicating that the error between $Q^\pi$ and $\tilde{Q}^\pi$ is orthogonal to $\nabla_{\theta_i} \log \pi$ for each $i$, and so cannot be reduced further by changing the parameters, at least locally. In this case equation (4) is unlikely to hold at a solution to (3), however with a good approximation architecture it may hold approximately, so that the we can derive an *estimate* of the Q-values from the policy using equation (5). We will use this estimate of the Q-values in the next section.

## 3.3  CONNECTION TO ACTION-VALUE METHODS

The previous section made a connection between regularized policy gradient and a regression onto the Q-values at the fixed point. In this section we go one step further, showing that actor-critic methods can be interpreted as action-value fitting methods, where the exact method depends on the choice of critic.

**Actor-critic methods.**  Consider an agent using an actor-critic method to learn both a policy $\pi$ and a value function $V$. At any iteration $k$, the value function $V^k$ has parameters $w^k$, and the policy is of the form

$$\pi^k(s,a) = \exp(W^k(s,a)/\alpha)/\sum_b \exp(W^k(s,b)/\alpha), \tag{8}$$

where $W^k$ is parameterized by $\theta^k$ and $\alpha > 0$ is the entropy regularization penalty. In this case $\nabla_\theta \log \pi^k(s,a) = (1/\alpha)(\nabla_\theta W^k(s,a) - \sum_b \pi(s,b)\nabla_\theta W^k(s,b))$. Using equation (6) the parameters are updated as

$$\Delta\theta \propto \underset{s,a}{\mathbf{E}}\,\delta_{\text{ac}}(\nabla_\theta W^k(s,a) - \sum_b \pi^k(s,b)\nabla_\theta W^k(s,b)), \quad \Delta w \propto \underset{s,a}{\mathbf{E}}\,\delta_{\text{ac}}\nabla_w V^k(s) \tag{9}$$

where $\delta_{\text{ac}}$ is the *critic minus baseline* term, which depends on the variant of actor-critic being used (see the remark below).

**Action-value methods.**  Compare this to the case where an agent is learning Q-values with a dueling architecture (Wang et al., 2016), which at iteration $k$ is given by

$$Q^k(s,a) = Y^k(s,a) - \sum_b \mu(s,b)Y^k(s,b) + V^k(s),$$

where $\mu$ is a probability distribution, $Y^k$ is parameterized by $\theta^k$, $V^k$ is parameterized by $w^k$, and the exploration policy is Boltzmann with temperature $\alpha$, *i.e.*,

$$\pi^k(s,a) = \exp(Y^k(s,a)/\alpha)/\sum_b \exp(Y^k(s,b)/\alpha). \tag{10}$$

In action value fitting methods at each iteration the parameters are updated to reduce some error, where the update is given by

$$\Delta\theta \propto \underset{s,a}{\mathbf{E}}\,\delta_{\text{av}}(\nabla_\theta Y^k(s,a) - \sum_b \mu(s,b)\nabla_\theta Y^k(s,b)), \quad \Delta w \propto \underset{s,a}{\mathbf{E}}\,\delta_{\text{av}}\nabla_w V^k(s) \tag{11}$$

where $\delta_{\text{av}}$ is the *action-value error* term and depends on which algorithm is being used (see the remark below).

**Equivalence.** The two policies (8) and (10) are identical if $W^k = Y^k$ for all $k$. Since $X^0$ and $Y^0$ can be initialized and parameterized in the same way, and assuming the two value function estimates are initialized and parameterized in the same way, all that remains is to show that the updates in equations (11) and (9) are identical. Comparing the two, and assuming that $\delta_{\mathrm{ac}} = \delta_{\mathrm{av}}$ (see remark), we see that the only difference is that the measure is not fixed in (9), but is equal to the current policy and therefore changes after each update. Replacing $\mu$ in (11) with $\pi^k$ makes the updates identical, in which case $W^k = Y^k$ at all iterations and the two policies (8) and (10) are always the same. In other words, the slightly modified action-value method is equivalent to an actor-critic policy gradient method, and vice-versa (modulo using the non-discounted distribution of states, as discussed in §2.2). In particular, regularized policy gradient methods can be interpreted as advantage function learning techniques (Baird III, 1993), since at the optimum the quantity $W(s, a) - \sum_b \pi(s, b)W(s, b) = \alpha(\log \pi(s, a) + H^\pi(s))$ will be equal to the advantage function values in the tabular case.

**Remark.** In SARSA (Rummery & Niranjan, 1994) we set $\delta_{\mathrm{av}} = r(s, a) + \gamma Q(s', b) - Q(s, a)$, where $b$ is the action selected at state $s'$, which would be equivalent to using a bootstrap critic in equation (6) where $Q^\pi(s, a) = r(s, a) + \gamma \tilde{Q}(s', b)$. In expected-SARSA (Sutton & Barto, 1998, Exercise 6.10), (Van Seijen et al., 2009)) we take the expectation over the Q-values at the next state, so $\delta_{\mathrm{av}} = r(s, a) + \gamma V(s') - Q(s, a)$. This is equivalent to TD-actor-critic (Konda & Tsitsiklis, 2003) where we use the value function to provide the critic, which is given by $Q^\pi = r(s, a) + \gamma V(s')$. In Q-learning (Watkins, 1989) $\delta_{\mathrm{av}} = r(s, a) + \gamma \max_b Q(s', b) - Q(s, a)$, which would be equivalent to using an optimizing critic that bootstraps using the max Q-value at the next state, *i.e.*, $Q^\pi(s, a) = r(s, a) + \gamma \max_b \tilde{Q}^\pi(s', b)$. In REINFORCE the critic is the Monte Carlo return from that state on, *i.e.*, $Q^\pi(s, a) = (\sum_{t=0}^\infty \gamma^t r_t \mid s_0 = s, a_0 = a)$. If the return trace is truncated and a bootstrap is performed after $n$-steps, this is equivalent to $n$-step SARSA or $n$-step Q-learning, depending on the form of the bootstrap (Peng & Williams, 1996).

## 3.4 BELLMAN RESIDUAL

In this section we show that $\|\mathcal{T}^\star Q^{\pi_\alpha} - Q^{\pi_\alpha}\| \to 0$ with decreasing regularization penalty $\alpha$, where $\pi_\alpha$ is the policy defined by (4) and $Q^{\pi_\alpha}$ is the corresponding Q-value function, both of which are functions of $\alpha$. We shall show that it converges to zero by bounding the sequence below by zero and above with a sequence that converges to zero. First, we have that $\mathcal{T}^\star Q^{\pi_\alpha} \geq \mathcal{T}^{\pi_\alpha} Q^{\pi_\alpha} = Q^{\pi_\alpha}$, since $\mathcal{T}^\star$ is greedy with respect to the Q-values. So $\mathcal{T}^\star Q^{\pi_\alpha} - Q^{\pi_\alpha} \geq 0$. Now, to bound from above we need the fact that $\pi_\alpha(s, a) = \exp(Q^{\pi_\alpha}(s, a)/\alpha) / \sum_b \exp(Q^{\pi_\alpha}(s, b)/\alpha) \leq \exp((Q^{\pi_\alpha}(s, a) - \max_c Q^{\pi_\alpha}(s, c))/\alpha)$. Using this we have

$$
\begin{aligned}
0 &\leq \mathcal{T}^\star Q^{\pi_\alpha}(s, a) - Q^{\pi_\alpha}(s, a) \\
&= \mathcal{T}^\star Q^{\pi_\alpha}(s, a) - \mathcal{T}^{\pi_\alpha} Q^{\pi_\alpha}(s, a) \\
&= \mathbf{E}_{s'}(\max_c Q^{\pi_\alpha}(s', c) - \sum_b \pi_\alpha(s', b) Q^{\pi_\alpha}(s', b)) \\
&= \mathbf{E}_{s'} \sum_b \pi_\alpha(s', b)(\max_c Q^{\pi_\alpha}(s', c) - Q^{\pi_\alpha}(s', b)) \\
&\leq \mathbf{E}_{s'} \sum_b \exp((Q^{\pi_\alpha}(s', b) - Q^{\pi_\alpha}(s', b^\star))/\alpha)(\max_c Q^{\pi_\alpha}(s', c) - Q^{\pi_\alpha}(s', b)) \\
&= \mathbf{E}_{s'} \sum_b f_\alpha(\max_c Q^{\pi_\alpha}(s', c) - Q^{\pi_\alpha}(s', b)),
\end{aligned}
$$

where we define $f_\alpha(x) = x \exp(-x/\alpha)$. To conclude our proof we use the fact that $f_\alpha(x) \leq \sup_x f_\alpha(x) = f_\alpha(\alpha) = \alpha \mathrm{e}^{-1}$, which yields

$$
0 \leq \mathcal{T}^\star Q^{\pi_\alpha}(s, a) - Q^{\pi_\alpha}(s, a) \leq |\mathcal{A}| \alpha \mathrm{e}^{-1}
$$

for all $(s, a)$, and so the Bellman residual converges to zero with decreasing $\alpha$. In other words, for small enough $\alpha$ (which is the regime we are interested in) the Q-values induced by the policy (4) will have a small Bellman residual. Moreover, this implies that $\lim_{\alpha \to 0} Q^{\pi_\alpha} = Q^\star$, as one might expect.

## 4 PGQL

In this section we introduce the main contribution of the paper, which is a technique to combine policy gradient with Q-learning. We call our technique 'PGQL', for policy gradient and Q-learning. In the previous section we showed that the Bellman residual is small at the fixed point of a regularized

policy gradient algorithm when the regularization penalty is sufficiently small. This suggests adding an auxiliary update where we explicitly attempt to reduce the Bellman residual as estimated from the policy, *i.e.*, a hybrid between policy gradient and Q-learning.

We first present the technique in a batch update setting, with a perfect knowledge of $Q^\pi$ (*i.e.*, a perfect critic). Later we discuss the practical implementation of the technique in a reinforcement learning setting with function approximation, where the agent generates experience from interacting with the environment and needs to estimate a critic simultaneously with the policy.

## 4.1 PGQL UPDATE

Define the estimate of $Q$ using the policy as

$$\tilde{Q}^\pi(s,a) = \alpha(\log \pi(s,a) + H^\pi(s)) + V(s), \tag{12}$$

where $V$ has parameters $w$ and is not necessarily $V^\pi$ as it was in equation (5). In (2) it was unnecessary to estimate the constant since the update was invariant to constant offsets, although in practice it is often estimated for use in a variance reduction technique (Williams, 1992; Sutton et al., 1999).

Since we know that at the fixed point the Bellman residual will be small for small $\alpha$, we can consider updating the parameters to reduce the Bellman residual in a fashion similar to Q-learning, *i.e.*,

$$\Delta\theta \propto \mathop{\mathbf{E}}_{s,a}(\mathcal{T}^\star\tilde{Q}^\pi(s,a) - \tilde{Q}^\pi(s,a))\nabla_\theta \log \pi(s,a), \quad \Delta w \propto \mathop{\mathbf{E}}_{s,a}(\mathcal{T}^\star\tilde{Q}^\pi(s,a) - \tilde{Q}^\pi(s,a))\nabla_w V(s).$$
$$\tag{13}$$

This is Q-learning applied to a particular form of the Q-values, and can also be interpreted as an actor-critic algorithm with an optimizing (and therefore biased) critic.

The full scheme simply combines two updates to the policy, the regularized policy gradient update (2) and the Q-learning update (13). Assuming we have an architecture that provides a policy $\pi$, a value function estimate $V$, and an action-value critic $Q^\pi$, then the parameter updates can be written as (suppressing the $(s,a)$ notation)

$$\Delta\theta \propto (1-\eta)\,\mathbf{E}_{s,a}(Q^\pi - \tilde{Q}^\pi)\nabla_\theta \log \pi + \eta\,\mathbf{E}_{s,a}(\mathcal{T}^\star\tilde{Q}^\pi - \tilde{Q}^\pi)\nabla_\theta \log \pi,$$
$$\Delta w \propto (1-\eta)\,\mathbf{E}_{s,a}(Q^\pi - \tilde{Q}^\pi)\nabla_w V + \eta\,\mathbf{E}_{s,a}(\mathcal{T}^\star\tilde{Q}^\pi - \tilde{Q}^\pi)\nabla_w V, \tag{14}$$

here $\eta \in [0,1]$ is a weighting parameter that controls how much of each update we apply. In the case where $\eta = 0$ the above scheme reduces to entropy regularized policy gradient. If $\eta = 1$ then it becomes a variant of (batch) Q-learning with an architecture similar to the dueling architecture (Wang et al., 2016). Intermediate values of $\eta$ produce a hybrid between the two. Examining the update we see that two error terms are trading off. The first term encourages consistency with critic, and the second term encourages optimality over time. However, since we know that under standard policy gradient the Bellman residual will be small, then it follows that adding a term that reduces that error should not make much difference at the fixed point. That is, the updates should be complementary, pointing in the same general direction, at least far away from a fixed point. This update can also be interpreted as an actor-critic update where the critic is given by a weighted combination of a standard critic and an optimizing critic. Yet another interpretation of the update is a combination of expected-SARSA and Q-learning, where the Q-values are parameterized as the sum of an advantage function and a value function.

## 4.2 PRACTICAL IMPLEMENTATION

The updates presented in (14) are batch updates, with an exact critic $Q^\pi$. In practice we want to run this scheme online, with an estimate of the critic, where we don't necessarily apply the policy gradient update at the same time or from same data source as the Q-learning update.

Our proposal scheme is as follows. One or more agents interact with an environment, encountering states and rewards and performing on-policy updates of (shared) parameters using an actor-critic algorithm where both the policy and the critic are being updated online. Each time an agent receives new data from the environment it writes it to a shared replay memory buffer. Periodically a separate learner process samples from the replay buffer and performs a step of Q-learning on the parameters of the policy using (13). This scheme has several advantages. The critic can accumulate the Monte

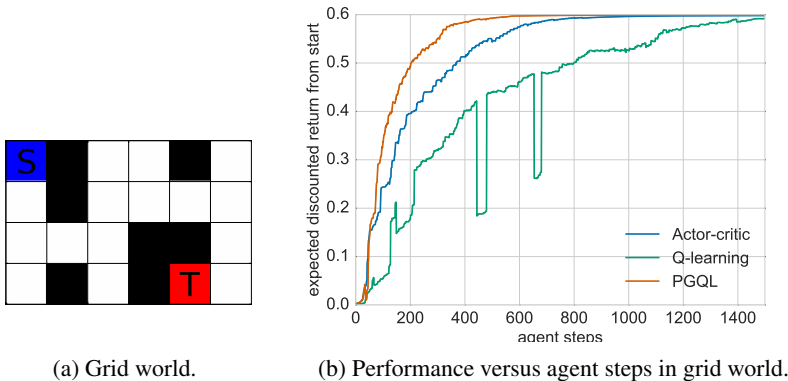

(a) Grid world.                    (b) Performance versus agent steps in grid world.

Figure 1: Grid world experiment.

Carlo return over many time periods, allowing us to spread the influence of a reward received in the future backwards in time. Furthermore, the replay buffer can be used to store and replay 'important' past experiences by prioritizing those samples (Schaul et al., 2015). The use of the replay buffer can help to reduce problems associated with correlated training data, as generated by an agent exploring an environment where the states are likely to be similar from one time step to the next. Also the use of replay can act as a kind of regularizer, preventing the policy from moving too far from satisfying the Bellman equation, thereby improving stability, in a similar sense to that of a policy 'trust-region' (Schulman et al., 2015). Moreover, by batching up replay samples to update the network we can leverage GPUs to perform the updates quickly, this is in comparison to pure policy gradient techniques which are generally implemented on CPU (Mnih et al., 2016).

Since we perform Q-learning using samples from a replay buffer that were generated by a old policy we are performing (slightly) off-policy learning. However, Q-learning is known to converge to the optimal Q-values in the off-policy tabular case (under certain conditions) (Sutton & Barto, 1998), and has shown good performance off-policy in the function approximation case (Mnih et al., 2013).

### 4.3  MODIFIED FIXED POINT

The PGQL updates in equation (14) have modified the fixed point of the algorithm, so the analysis of §3 is no longer valid. Considering the tabular case once again, it is still the case that the policy $\pi \propto \exp(\tilde{Q}^\pi/\alpha)$ as before, where $\tilde{Q}^\pi$ is defined by (12), however where previously the fixed point satisfied $\tilde{Q}^\pi = Q^\pi$, with $Q^\pi$ corresponding to the Q-values induced by $\pi$, now we have

$$\tilde{Q}^\pi = (1-\eta)Q^\pi + \eta\mathcal{T}^\star\tilde{Q}^\pi, \tag{15}$$

Or equivalently, if $\eta < 1$, we have $\tilde{Q}^\pi = (1-\eta)\sum_{k=0}^\infty \eta^k(\mathcal{T}^\star)^k Q^\pi$. In the appendix we show that $\|\tilde{Q}^\pi - Q^\pi\| \to 0$ and that $\|\mathcal{T}^\star Q^\pi - Q^\pi\| \to 0$ with decreasing $\alpha$ in the tabular case. That is, for small $\alpha$ the induced Q-values and the Q-values estimated from the policy are close, and we still have the guarantee that in the limit the Q-values are optimal. In other words, we have not perturbed the policy very much by the addition of the auxiliary update.

## 5  NUMERICAL EXPERIMENTS

### 5.1  GRID WORLD

In this section we discuss the results of running PGQL on a toy 4 by 6 grid world, as shown in Figure 1a. The agent always begins in the square marked 'S' and the episode continues until it reaches the square marked 'T', upon which it receives a reward of 1. All other times it receives no reward. For this experiment we chose regularization parameter $\alpha = 0.001$ and discount factor $\gamma = 0.95$.

Figure 1b shows the performance traces of three different agents learning in the grid world, running from the same initial random seed. The lines show the *true* expected performance of the policy

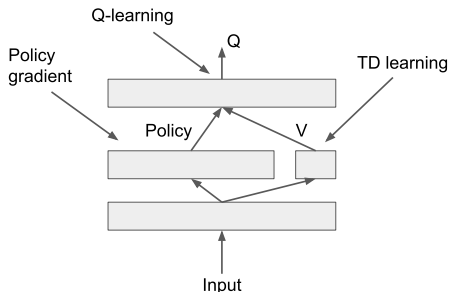

Figure 2: PGQL network augmentation.

from the start state, as calculated by value iteration after each update. The blue-line is standard TD-actor-critic (Konda & Tsitsiklis, 2003), where we maintain an estimate of the value function and use that to generate an estimate of the Q-values for use as the critic. The green line is Q-learning where at each step an update is performed using data drawn from a replay buffer of prior experience and where the Q-values are parameterized as in equation (12). The policy is a softmax over the Q-value estimates with temperature $\alpha$. The red line is PGQL, which at each step first performs the TD-actor-critic update, then performs the Q-learning update as in (14).

The grid world was totally deterministic, so the step size could be large and was chosen to be $1$. A step-size any larger than this made the pure actor-critic agent fail to learn, but both PGQL and Q-learning could handle some increase in the step-size, possibly due to the stabilizing effect of using replay.

It is clear that PGQL outperforms the other two. At any point along the x-axis the agents have seen the same amount of data, which would indicate that PGQL is more data efficient than either of the vanilla methods since it has the highest performance at practically every point.

## 5.2  ATARI

We tested our algorithm on the full suite of Atari benchmarks (Bellemare et al., 2012), using a neural network to parameterize the policy. In figure 2 we show how a policy network can be augmented with a parameterless additional layer which outputs the Q-value estimate. With the exception of the extra layer, the architecture and parameters were chosen to exactly match the asynchronous advantage actor-critic (A3C) algorithm presented in Mnih et al. (2016), which in turn reused many of the settings from Mnih et al. (2015). Specifically we used the exact same learning rate, number of workers, entropy penalty, bootstrap horizon, and network architecture. This allows a fair comparison between A3C and PGQL, since the only difference is the addition of the Q-learning step. Our technique augmented A3C with the following change: After each actor-learner has accumulated the gradient for the policy update, it performs a single step of Q-learning from replay data as described in equation (13), where the minibatch size was 32 and the Q-learning learning rate was chosen to be $0.5$ times the actor-critic learning rate (we mention learning rate ratios rather than choice of $\eta$ in (14) because the updates happen at different frequencies and from different data sources). Each actor-learner thread maintained a replay buffer of the last $100k$ transitions seen by that thread. We ran the learning for $50$ million agent steps ($200$ million Atari frames), as in (Mnih et al., 2016).

In the results we compare against both A3C and a variant of asynchronous deep Q-learning. The changes we made to Q-learning are to make it similar to our method, with some tuning of the hyper-parameters for performance. We use the exact same network, the exploration policy is a softmax over the Q-values with a temperature of $0.1$, and the Q-values are parameterized as in equation (12) (*i.e.*, similar to the dueling architecture (Wang et al., 2016)), where $\alpha = 0.1$. The Q-value updates are performed every 4 steps with a minibatch of 32 (roughly 5 times more frequently than PGQL). For each method, all games used identical hyper-parameters.

The results across all games are given in table 3 in the appendix. All scores have been normalized by subtracting the average score achieved by an agent that takes actions uniformly at random. Each game was tested 5 times per method with the same hyper-parameters but with different ran-

dom seeds. The scores presented correspond to the best score obtained by any run from a random start evaluation condition (Mnih et al., 2016). Overall, PGQL performed best in 34 games, A3C performed best in 7 games, and Q-learning was best in 10 games. In 6 games two or more methods tied. In tables 1 and 2 we give the mean and median normalized scores as percentage of an expert human normalized score across all games for each tested algorithm from random and human-start conditions respectively. In a human-start condition the agent takes over control of the game from randomly selected human-play starting points, which generally leads to lower performance since the agent may not have found itself in that state during training. In both cases, PGQL has both the highest mean and median, and the median score exceeds 100%, the human performance threshold.

It is worth noting that PGQL was the worst performer in only one game, in cases where it was not the outright winner it was generally somewhere in between the performance of the other two algorithms. Figure 3 shows some sample traces of games where PGQL was the best performer. In these cases PGQL has far better data efficiency than the other methods. In figure 4 we show some of the games where PGQL under-performed. In practically every case where PGQL did not perform well it had better data efficiency early on in the learning, but performance saturated or collapsed. We hypothesize that in these cases the policy has reached a local optimum, or over-fit to the early data, and might perform better were the hyper-parameters to be tuned.

|  | A3C | Q-learning | PGQL |
|---|---|---|---|
| **Mean** | 636.8 | 756.3 | 877.2 |
| **Median** | 107.3 | 58.9 | 145.6 |

Table 1: Mean and median normalized scores for the Atari suite from random starts, as a percentage of human normalized score.

|  | A3C | Q-learning | PGQL |
|---|---|---|---|
| **Mean** | 266.6 | 246.6 | 416.7 |
| **Median** | 58.3 | 30.5 | 103.3 |

Table 2: Mean and median normalized scores for the Atari suite from human starts, as a percentage of human normalized score.

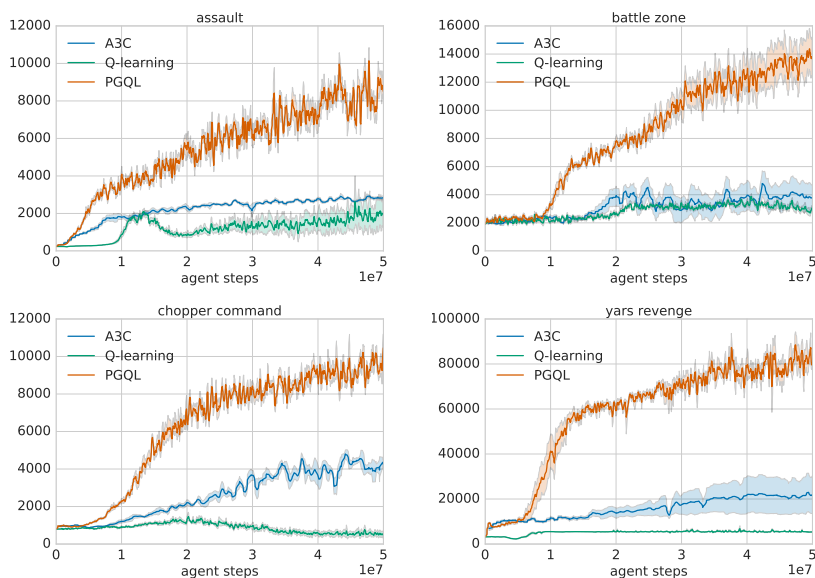

Figure 3: Some Atari runs where PGQL performed well.

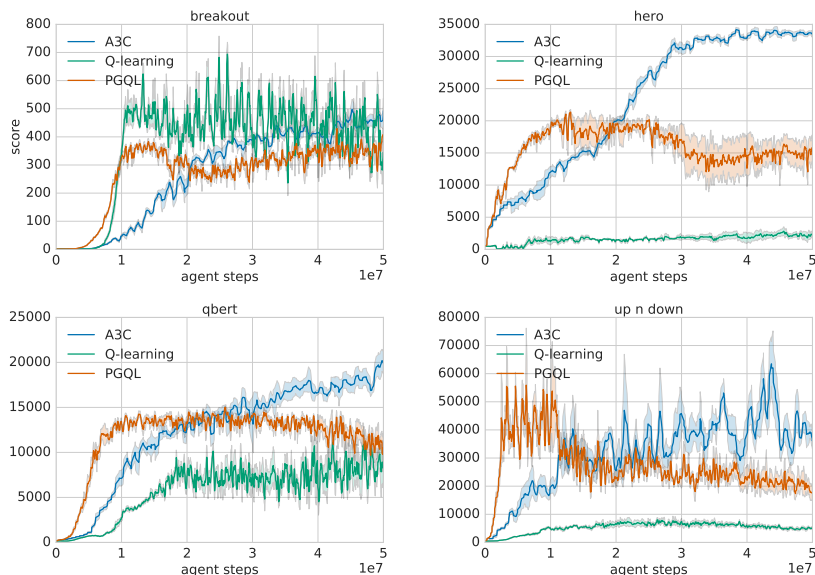

Figure 4: Some Atari runs where PGQL performed poorly.

## 6 CONCLUSIONS

We have made a connection between the fixed point of regularized policy gradient techniques and the Q-values of the resulting policy. For small regularization (the usual case) we have shown that the Bellman residual of the induced Q-values must be small. This leads us to consider adding an auxiliary update to the policy gradient which is related to the Bellman residual evaluated on a transformation of the policy. This update can be performed off-policy, using stored experience. We call the resulting method 'PGQL', for policy gradient and Q-learning. Empirically, we observe better data efficiency and stability of PGQL when compared to actor-critic or Q-learning alone. We verified the performance of PGQL on a suite of Atari games, where we parameterize the policy using a neural network, and achieved performance exceeding that of both A3C and Q-learning.

## 7 ACKNOWLEDGMENTS

We thank Joseph Modayil for many comments and suggestions on the paper, and Hubert Soyer for help with performance evaluation. We would also like to thank the anonymous reviewers for their constructive feedback.

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

# A  PGQL BELLMAN RESIDUAL

Here we demonstrate that in the tabular case the Bellman residual of the induced Q-values for the PGQL updates of (14) converges to zero as the temperature $\alpha$ decreases, which is the same guarantee as vanilla regularized policy gradient (2). We will use the notation that $\pi_\alpha$ is the policy at the fixed point of PGQL updates (14) for some $\alpha$, *i.e.*, $\pi_\alpha \propto \exp(\tilde{Q}^{\pi_\alpha})$, with induced Q-value function $Q^{\pi_\alpha}$.

First, note that we can apply the same argument as in §3.4 to show that $\lim_{\alpha \to 0} \|\mathcal{T}^\star \tilde{Q}^{\pi_\alpha} - \mathcal{T}^{\pi_\alpha} \tilde{Q}^{\pi_\alpha}\| = 0$ (the only difference is that we lack the property that $\tilde{Q}^{\pi_\alpha}$ is the fixed point of $\mathcal{T}^{\pi_\alpha}$). Secondly, from equation (15) we can write $\tilde{Q}^{\pi_\alpha} - Q^{\pi_\alpha} = \eta(\mathcal{T}^\star \tilde{Q}^{\pi_\alpha} - Q^{\pi_\alpha})$. Combining these two facts we have

$$
\begin{aligned}
\|\tilde{Q}^{\pi_\alpha} - Q^{\pi_\alpha}\| &= \eta \|\mathcal{T}^\star \tilde{Q}^{\pi_\alpha} - Q^{\pi_\alpha}\| \\
&= \eta \|\mathcal{T}^\star \tilde{Q}^{\pi_\alpha} - \mathcal{T}^{\pi_\alpha} \tilde{Q}^{\pi_\alpha} + \mathcal{T}^{\pi_\alpha} \tilde{Q}^{\pi_\alpha} - Q^{\pi_\alpha}\| \\
&\leq \eta(\|\mathcal{T}^\star \tilde{Q}^{\pi_\alpha} - \mathcal{T}^{\pi_\alpha} \tilde{Q}^{\pi_\alpha}\| + \|\mathcal{T}^{\pi_\alpha} \tilde{Q}^{\pi_\alpha} - \mathcal{T}^{\pi_\alpha} Q^{\pi_\alpha}\|) \\
&\leq \eta(\|\mathcal{T}^\star \tilde{Q}^{\pi_\alpha} - \mathcal{T}^{\pi_\alpha} \tilde{Q}^{\pi_\alpha}\| + \gamma \|\tilde{Q}^{\pi_\alpha} - Q^{\pi_\alpha}\|) \\
&\leq \eta/(1 - \eta\gamma) \|\mathcal{T}^\star \tilde{Q}^{\pi_\alpha} - \mathcal{T}^{\pi_\alpha} \tilde{Q}^{\pi_\alpha}\|,
\end{aligned}
$$

and so $\|\tilde{Q}^{\pi_\alpha} - Q^{\pi_\alpha}\| \to 0$ as $\alpha \to 0$. Using this fact we have

$$
\begin{aligned}
\|\mathcal{T}^\star \tilde{Q}^{\pi_\alpha} - \tilde{Q}^{\pi_\alpha}\| &= \|\mathcal{T}^\star \tilde{Q}^{\pi_\alpha} - \mathcal{T}^{\pi_\alpha} \tilde{Q}^{\pi_\alpha} + \mathcal{T}^{\pi_\alpha} \tilde{Q}^{\pi_\alpha} - Q^{\pi_\alpha} + Q^{\pi_\alpha} - \tilde{Q}^{\pi_\alpha}\| \\
&\leq \|\mathcal{T}^\star \tilde{Q}^{\pi_\alpha} - \mathcal{T}^{\pi_\alpha} \tilde{Q}^{\pi_\alpha}\| + \|\mathcal{T}^{\pi_\alpha} \tilde{Q}^{\pi_\alpha} - \mathcal{T}^{\pi_\alpha} Q^{\pi_\alpha}\| + \|Q^{\pi_\alpha} - \tilde{Q}^{\pi_\alpha}\| \\
&\leq \|\mathcal{T}^\star \tilde{Q}^{\pi_\alpha} - \mathcal{T}^{\pi_\alpha} \tilde{Q}^{\pi_\alpha}\| + (1 + \gamma) \|\tilde{Q}^{\pi_\alpha} - Q^{\pi_\alpha}\| \\
&< 3/(1 - \eta\gamma) \|\mathcal{T}^\star \tilde{Q}^{\pi_\alpha} - \mathcal{T}^{\pi_\alpha} \tilde{Q}^{\pi_\alpha}\|,
\end{aligned}
$$

which therefore also converges to zero in the limit. Finally we obtain

$$
\begin{aligned}
\|\mathcal{T}^\star Q^{\pi_\alpha} - Q^{\pi_\alpha}\| &= \|\mathcal{T}^\star Q^{\pi_\alpha} - \mathcal{T}^\star \tilde{Q}^{\pi_\alpha} + \mathcal{T}^\star \tilde{Q}^{\pi_\alpha} - \tilde{Q}^{\pi_\alpha} + \tilde{Q}^{\pi_\alpha} - Q^{\pi_\alpha}\| \\
&\leq \|\mathcal{T}^\star Q^{\pi_\alpha} - \mathcal{T}^\star \tilde{Q}^{\pi_\alpha}\| + \|\mathcal{T}^\star \tilde{Q}^{\pi_\alpha} - \tilde{Q}^{\pi_\alpha}\| + \|\tilde{Q}^{\pi_\alpha} - Q^{\pi_\alpha}\| \\
&\leq (1 + \gamma) \|\tilde{Q}^{\pi_\alpha} - Q^{\pi_\alpha}\| + \|\mathcal{T}^\star \tilde{Q}^{\pi_\alpha} - \tilde{Q}^{\pi_\alpha}\|,
\end{aligned}
$$

which combined with the two previous results implies that $\lim_{\alpha \to 0} \|\mathcal{T}^\star Q^{\pi_\alpha} - Q^{\pi_\alpha}\| = 0$, as before.

# B  Atari scores

| Game | A3C | Q-learning | PGQL |
|---|---|---|---|
| alien | 38.43 | 25.53 | **46.70** |
| amidar | 68.69 | 12.29 | **71.00** |
| assault | 854.64 | 1695.21 | **2802.87** |
| asterix | 191.69 | 98.53 | **3790.08** |
| asteroids | 24.37 | 5.32 | **50.23** |
| atlantis | 15496.01 | 13635.88 | **16217.49** |
| bank heist | 210.28 | 91.80 | **212.15** |
| battle zone | 21.63 | 2.89 | **52.00** |
| beam rider | 59.55 | 79.94 | **155.71** |
| berzerk | 79.38 | 55.55 | **92.85** |
| bowling | 2.70 | -7.09 | **3.85** |
| boxing | 510.30 | 299.49 | **902.77** |
| breakout | 2341.13 | **3291.22** | 2959.16 |
| centipede | 50.22 | **105.98** | 73.88 |
| chopper command | 61.13 | 19.18 | **162.93** |
| crazy climber | **510.25** | 189.01 | 476.11 |
| defender | 475.93 | 58.94 | **911.13** |
| demon attack | **4027.57** | 3449.27 | 3994.49 |
| double dunk | 1250.00 | 91.35 | **1375.00** |
| enduro | **9.94** | **9.94** | **9.94** |
| fishing derby | 140.84 | -14.48 | **145.57** |
| freeway | -0.26 | **-0.13** | **-0.13** |
| frostbite | 5.85 | **10.71** | 5.71 |
| gopher | 429.76 | **9131.97** | 2060.41 |
| gravitar | 0.71 | 1.35 | **1.74** |
| hero | **145.71** | 15.47 | 92.88 |
| ice hockey | 62.25 | 21.57 | **76.96** |
| jamesbond | 133.90 | 110.97 | **142.08** |
| kangaroo | -0.94 | -0.94 | **-0.75** |
| krull | 736.30 | **3586.30** | 557.44 |
| kung fu master | 182.34 | **260.14** | 254.42 |
| montezuma revenge | -0.49 | **1.80** | -0.48 |
| ms pacman | 17.91 | 10.71 | **25.76** |
| name this game | 102.01 | 113.89 | **188.90** |
| phoenix | 447.05 | 812.99 | **1507.07** |
| pitfall | 5.48 | **5.49** | **5.49** |
| pong | **116.37** | 24.96 | **116.37** |
| private eye | -0.88 | **0.03** | -0.04 |
| qbert | **186.91** | 159.71 | 136.17 |
| riverraid | 107.25 | 65.01 | **128.63** |
| road runner | **603.11** | 179.69 | 519.51 |
| robotank | 15.71 | **134.87** | 71.50 |
| seaquest | 3.81 | 3.71 | **5.88** |
| skiing | **54.27** | 54.10 | 54.16 |
| solaris | 27.05 | **34.61** | 28.66 |
| space invaders | 188.65 | 146.39 | **608.44** |
| star gunner | 756.60 | 205.70 | **977.99** |
| surround | 28.29 | -1.51 | **78.15** |
| tennis | **145.58** | -15.35 | **145.58** |
| time pilot | 270.74 | 91.59 | **438.50** |
| tutankham | 224.76 | 110.11 | **239.58** |
| up n down | **1637.01** | 148.10 | 1484.43 |
| venture | **-1.76** | **-1.76** | **-1.76** |
| video pinball | 3007.37 | 4325.02 | **4743.68** |
| wizard of wor | 150.52 | 88.07 | **325.39** |
| yars revenge | 81.54 | 23.39 | **252.83** |
| zaxxon | 4.01 | 44.11 | **224.89** |

Table 3: Normalized scores for the Atari suite from random starts, as a percentage of human normalized score.

