# Peer review of "Combining policy gradient and Q-learning"

_ICLR 2017 — accepted_

[Public Comment · Tara N Sainath · 07 Nov 2016]
**ICLR Paper Format**

Dear Authors,

Please resubmit your paper in the ICLR 2017 format with the correct margin spacing for your submission to be considered. Thank you!

[Reviewer Comment · AnonReviewer1 · 05 Dec 2016]
**Independence assumption**

Hi,

In 3.2 you consider the case where "the measure in the expectation (is) independent of theta". However this is not the case in practice since both the state and action distributions depend on theta. Could you please comment on how this affects the proposed interpretation? I believe it would be important to explain it in the paper.

Thanks!

[Reviewer Comment · AnonReviewer1 · 08 Dec 2016]
**A few more questions**

Hi,

Could you please clarify the following:
  1. It sounds like alpha is kept fixed in experiments and results are reported for the "exploratory" policy (softmax with temperature alpha) rather than the greedy one. Is that correct and if yes, why?
  2. In both the grid world and Atari, is the critic estimate of Q(s, a) obtained by summing the (discounted) observed rewards for up to t_max timesteps after taking action a in state s, plus the (discounted) estimated V(last observed state)? (= as in A3C)
  3. For the Q-learning step are you freezing the target network and clipping the error as in the Nature DQN paper? If not, why?

Thanks!

[Public Comment · (anonymous) · 09 Dec 2016]
**Page 6, last paragraph (a small doubt)**

The authors write at the page 6, last paragraph "Under standard policy gradient the bellman residual will be small, then it follows that adding a term that reduces that error should not make much difference at the fixed point." 

This point is not clear. Let's consider the standard policy gradient without the entropy-bonus. In this case, I was wondering if there is a way to determine the fixed-policy? I did the same maths inspired by the paper but I did not reach to any concluding result.

[Reviewer Comment · AnonReviewer1 · 10 Dec 2016]
**Discounted weighting of states?**

When initially reading the paper I assumed you just forgot to mention in section 2 that the distribution d^pi used a discounted weighted of states, for the policy gradient theorem from Sutton et al (1999) to hold (see above eq. 2 in that paper), however it seems like no such discounting is being applied when actually doing the updates in practice. Obviously in the tabular case it does not matter, but could it be a problem with DQNs? (which value of gamma is being used by the way?)

[Official Review · AnonReviewer3 · rating 7 · confidence 3 · 17 Dec 2016]

Nice paper, exploring the connection between value-based methods and policy gradients, formalizing the relation between the softmax-like policy induced by the Q-values and a regularized form of PG.  

Presentation: 
Although that seems to be the flow in the first part of the paper, I think it could be cast as a extension/ generalization of the dueling Q-network – for me that would be a more intuitive exposition of the new algorithm and findings. 

Small concern in general case derivation: 
Section 3.2: Eq. (7) the expectation (s,a) is wrt to \pi, which is a function of \theta -- that dependency seems to be ignored, although it is key to the PG update derivation. If these policies(the sampling policy for the expectation and \pi) are close enough it's usually okay -- but except for particular cases (trust-region methods & co), that's generally not true. Thus, you might end up solving a very different problem than the one you actually care solving.

Results:
A comparison with the dueling architecture could be added as that would be the closest method (it would be nice to see if and in which game you get an improvement)

Overall: strong paper, good theoretical insights.

[Official Review · AnonReviewer1 · rating 7 · confidence 4 · 17 Dec 2016]
**Interesting links between policy-based and value-based methods**

This paper shows how policy gradient and Q-Learning may be combined together, improving learning as demonstrated in particular in the Atari Learning Environment. The core idea is to note that entropy-regularized policy gradient leads to a Boltzmann policy based on Q values, thus linking pi & Q together and allowing both policy gradient and Q-Learning updates to be applied.

I think this is a very interesting paper, not just for its results and the proposed algorithm (dubbed PGQ), but mostly because of the links it draws between several techniques, which I found quite insightful.

That being said, I also believe it could have done a better job at clearly exposing these links: I found it somewhat difficult to follow, and it took me a while to wrap my head around it, even though the underlying concepts are not that complex. In particular:
- The notation \tilde{Q}^pi is introduced in a way that is not very clear, as "an estimate of the Q-values" while eq. 5 is an exact equality (no estimation)
- It is not clear to me what section 3.2 is bringing exactly, I wonder if it could just be removed to expand some other sections with more explanations.
- The links to dueling networks (Wang et al, 2016) are in my opinion not explicit and detailed enough (in 3.3 & 4.1): as far as I can tell the proposed architecture ends up being very similar to such networks and thus it would be worth telling more about it (also in experiments my understanding is that the "variant of asynchronous deep Q-learning" being used is essentially such a dueling network, but it is not clearly stated). I also believe it should be mentioned that PGQ can also be seen as combining Q-Learning with n-step expected Sarsa using a dueling network: this kind of example helps better understand the links between methods
- Overall I wish section 3.3 was clearer, as it draws some very interesting links, but it is hard to see where this is all going when reading the paper for the first time. One confusing point is w.r.t. to the relationship with section 3.2, that assumes a critic outputting Q values while in 3.3 the critic outputs V. The "mu" distribution also comes somewhat out of nowhere.

I hope the authors can try and improve the readability of the paper in a final version, as well as clarify the points raised in pre-review questions (in particular related to experimental details, the derivation of eq. 4, and the issue of the discounted distribution of states).

Minor remarks:
- The function r(s, a) used in the Bellman equation in section 2 is not formally defined. It looks a bit weird because the expectation is on s' and b' but r(s, a) does not depend on them (so either it should be moved out of the expectation, or the expectation should also be over the reward, depending on how r is defined)
- The definition of the Boltzmann policy at end of 2.1 is a bit confusing since there is a sum over "a" of a quantity that does not depend (clearly) on "a"
- I believe 4.3 is for the tabular case but this is not clearly stated
- Any idea why in Fig. 1 the 3 algorithms do not all converge to the same policy? In such a simple toy setting I would expect it to be the case.

Typos:
- "we refer to the classic text Sutton & Barto (1998)" => missing "by"?
- "Online policy gradient typically require an estimate of the action-values function" => requires & value
- "the agent generates experience from interacting the environment" => with the environment
- in eq. 12 (first line) there is a comma to remove near the end, just before the dlog pi
- "allowing us the spread the influence of a reward" => to spread
- "in the off-policy case tabular case" => remove the first case
- "The green line is Q-learning where at the step an update is performed" => at each step
- In Fig. 2 it says A2C instead of A3C

NB: I did not have time to carefully read Appendix A

[Official Review · AnonReviewer4 · rating 9 · confidence 5 · 27 Dec 2016]
**An excellent paper pointing out connections between existing algorithms, which also leads to new algorithms with excellent results**

This is a very nicely written paper which unifies some value-based and policy-based (regularized policy gradient) methods, by pointing out connections between the value function and policy which have not been established before. The theoretical results are new and insightful, and will likely be useful in the RL field much beyond the specific algorithm being proposed in the paper. This being said, the paper does exploit the theory to produce a unified version of Q-learning and policy gradient, which proves to work on par or better than the state-of-art algorithms on the Atari suite. The empirical section is very well explained in terms of what optimization were done.
One minor comment I had was related to the stationary distribution used for a policy - there are subtleties here between using a discounted vs non-discounted distribution which are not crucial in the tabular case, but will need to be addressed in the long run in the function approximation case. This being said, there is no major problem for the current version of the paper. 
Overall, the paper is definitely worthy of acceptance, and will likely influence a broad swath of RL, as it opens the door to further theoretical results as well as algorithm development.

[Public Comment · Brendan ODonoghue · 17 Jan 2017]
**Thankyou to the reviewers**

Thankyou to the three reviewers, and apologies for the delay in responding. We have updated the paper in response to your comments. Further responses below.

[Final Decision · Program Chairs · 06 Feb 2017]
**ICLR committee final decision**

Reviewers agree that this a high-quality and interesting paper exploring important connections between widely used RL algorithms. It has the potential to be an impactful paper, with the most positive comment noting that it "will likely influence a broad swath of RL". 
 
 Pros:
 - The main concepts of the paper came through clearly, and all reviewers felt the idea was interesting and novel.
 - The empirical part of the paper was convincing and "empirical section is very well explained" 
 
 Cons:
 - There and some concerns about the writing and notation. None of these were too critical though, and the authors responded 
 - Reviewers asked for some more comparisons with alternative formulations.